# Built environment profiles for Latin American urban settings: The SALURBAL study

Olga L. Sarmiento[1]*, Andrés F. Useche[2], Daniel A. Rodriguez[3], Iryna Dronova[4], Oscar Guaje[2], Felipe Montes[2], Ivana Stankov[5,6], Maria Alejandra Wilches[2], Usama Bilal[5,6], Xize Wang[7], Luis A. Guzmán[8], Fabian Peña[9], D. Alex Quistberg[6,10], John A. Guerra-Gomez[9,11], Ana V. Diez Roux[5,6]

1 School of Medicine, Universidad de Los Andes in Bogotá Colombia, Bogotá, Colombia, 2 Department of Industrial Engineering, School of Engineering, Universidad de Los Andes in Bogotá Colombia, Bogotá, Colombia, 3 College of Environmental Design and Institute for Transportation Studies, University of California Berkeley, Berkeley, CA, United States of America, 4 Department of Landscape Architecture and Environmental Planning, University of California, Berkeley, Berkeley, CA, United States of America, 5 Department of Epidemiology and Biostatistics, Dornsife School of Public Health, Drexel University, Philadelphia, PA, United States of America, 6 Urban Health Collaborative, Dornsife School of Public Health, Drexel University, Philadelphia, PA, United States of America, 7 Department of Real Estate, National University of Singapore, Singapore, Singapore, 8 Department of Civil and Environmental Engineering, School of Engineering, Universidad de Los Andes in Bogotá Colombia, Bogotá, Colombia, 9 Department of Computer Science, School of Engineering, Universidad de Los Andes in Bogotá Colombia, Bogotá, Colombia, 10 Department of Environmental and Occupational Health, Dornsife School of Public Health, Drexel University, Philadelphia, PA, United States of America, 11 Khoury School of Computer Science, Northeastern University, San Jose, CA, United States of America

* osarmien@uniandes.edu.co

**Data Availability Statement:** All relevant data are within the paper and its Supporting Information files.

## Abstract

The built environment of cities is complex and influences social and environmental determinants of health. In this study we, 1) identified city profiles based on the built landscape and street design characteristics of cities in Latin America and 2) evaluated the associations of city profiles with social determinants of health and air pollution. Landscape and street design profiles of 370 cities were identified using finite mixture modeling. For landscape, we measured fragmentation, isolation, and shape. For street design, we measured street connectivity, street length, and directness. We fitted a two-level linear mixed model to assess the association of social and environmental determinants of health with the profiles. We identified four profiles for landscape and four for the street design domain. The most common landscape profile was the "proximate stones" characterized by moderate fragmentation, isolation and patch size, and irregular shape. The most common street design profile was the "semi-hyperbolic grid" characterized by moderate connectivity, street length, and directness. The "semi-hyperbolic grid", "spiderweb" and "hyperbolic grid" profiles were positively associated with higher access to piped water and less overcrowding. The "semi-hyperbolic grid" and "spiderweb" profiles were associated with higher air pollution. The "proximate stones" and "proximate inkblots" profiles were associated with higher congestion. In conclusion, there is substantial heterogeneity in the urban landscape and street design profiles of Latin American cities. While we did not find a specific built environment profile that was consistently associated with lower air pollution and better social conditions, the different

**Funding:** Olga L. Sarmiento, Andrés F. Useche, Daniel A. Rodriguez, Iryna Dronova, Oscar Guaje, Felipe Montes, Ivana Stankov, Maria Alejandra Wilches, Usama Bilal, Xize Wang, Luis A. Guzmán, Fabian Peña, D. Alex Quistberg, John A. Guerra-Gomez, and, Ana V. Diez Roux were funded by the Wellcome Trust [205177/Z/16/Z]. The Salud Urbana en América Latina (SALURBAL)/Urban Health in Latin America project was funded by the Wellcome Trust [205177/Z/16/Z]. In addition, Usama Bilal was also funded by the National Institutes of Health (NIH) under award number DP5OD26429. Mention of trade names, commercial practices, or organizations does not imply endorsement by the authors, the institutions where the authors work, or funding entities. The funders had no role in study design, data collection, and analysis, decision to publish, or preparation of the manuscript. There was no additional external funding received for this study. More information about the SALURBAL project can be found at https://drexel.edu/lac.

**Competing interests:** The authors have declared that no competing interests exist.

configurations of the built environments of cities should be considered when planning healthy and sustainable cities in Latin America.

## Introduction

Cities are highly complex systems in which dynamic networks between people and the built environment give rise to patterns in behaviors and health [1, 2]. Specifically, certain characteristics of the built environment have been associated with social and environmental determinants of health (e.g., poverty, transport, air pollution, access to public goods), mental health [3], non-communicable diseases (NCD) [4], road traffic deaths [5], and inequities in health [6]. Nonetheless, most descriptive information on city-built environments and social determinants of health has focused on high-income countries [7] or large cities from low-to-middle income countries; few studies have focused on Latin American cities [8–10].

Certain features of Latin American cities make an examination of the built environment of these cities of special interest. Latin America is dense and highly urbanized [11, 12] where around 80% of the population lives in urban areas [12]. The region is notoriously unequal [13], with 24% of the population living in dense informal settlements [14]. Congestion is a major challenge for many large Latin American cities [15]. In addition, 58% of the urban population lives in areas with air pollution levels above the defined WHO-AQG of 10 μg/m3 annual average [16]. Despite these challenges, Latin America is also known for its innovative and sustainable transport policies [17], urban development projects [18], and social programs [19].

In this context, understanding factors that explain the configuration of the built environment in Latin American cities may provide useful insights for urban policies. Factors that may impact the configuration of cities include historical legacies related to when the area was settled, how it was settled, and urban form indicators [20]. Specifically, Latin American cities that were Iberian colonies have more regular urban layouts and denser street networks [10]. Furthermore, for each city, the historical period of most intense growth may also matter, as growth during periods with higher private vehicle ownership may give rise to urban landscape and street design patterns that differ from those during earlier periods of growth. The physical geography of the area, including the terrain (e.g., cities built in the Andean mountains are less extensive, denser, and have a higher risk of natural disasters [10]), proximity to water bodies and arable lands, are also likely to impact the shape and characteristics of cities [21]. Policy factors contribute to shape and reshape urban spaces in the Latin American region [22]. These factors include incentives or disincentives to grow in certain areas (e.g., monocentric [Lima] vs. polycentric [Mexico city]), requirements concerning the characteristics of that growth (e.g., density, land uses, informal settlements with low integration with formal settlements), road infrastructure and transportation policies (e.g., concentration of public transport in central areas with limited access in the peripheries and suburban areas and road building), and economic development policies [23]. Finally, the patterns of urbanization, including, high levels of density compared to other regions, are in part a reflection of recent accelerated urbanization since the middle of the 20th century coupled with a high proportion of the population living in informal settlements [24].

In this context, it is crucial to better understand the heterogeneity of the built environment of Latin American cities [10]. The few studies of the built environment of Latin American cities have shown that compared to other cities in the world, Latin American cities are denser

and less complex in shape than cities in North America and Western Europe [7], but less dense than cities in sub-Saharan Africa and South-East Asia [25]. Overall, cities in South America and the Caribbean show a weak trend towards less-fragmented urban growth between 2000 and 2010 while Central American cities exhibit more fragmented urban growth [10]. Large cities in Latin America exhibit a trend towards increasing fragmentation and decreasing density [26]. Regarding street design, a previous study including 919 cities in Latin America showed that this region is characterized by high street and intersection density [10]. In contrast, the configuration of street networks across 118 urban areas in 12 metropolitan regions showed that US city street networks evolved to lower street density and less directed streets [27]. In addition, a study assessing street design metrics in 100 cities across 63 countries, including nine cities from eight Latin American countries showed that cities in Latin America have fewer directed streets than in cities in the US and Canada, but more directed streets than cities in other continents [28].

Several studies have been conducted to assess the associations of the built environment with social determinants of health and air pollution [29]. Social determinants of health are the conditions that people are born into and live under that affect their health. Currently, remote sensing imagery, and automatic data retrieval systems (such as OSMnx have enabled a scaling up of these types of studies to include multiple cities in the world. To date, however, only a few studies have included Latin American cities and assessed their relationship of city-built environments with social and environmental determinants of health [7]. A study with 77 cities from Europe, the US, Australia, and Asia and 12 capital cities from Latin America showed that higher country-level socioeconomic indicators were correlated positively with more complex landscapes and larger proportions of open space, and negatively with density and compactness [7]. This study also found a positive correlation between motorization rate and complex landscape, but the relationship of congestion and urban form has not been evaluated in Latin American cities [7]. A study including 919 cities in Latin American showed that compact and well-connected cities exhibit higher levels of productivity measured by density of radiance from the nighttime imagery [10]. A previous study of SALURBAL (Salud Urbana en América Latina or Urban Health in Latin America) including 366 cities in 11 countries from Latin America using satellite-derived PM measures showed that higher intersection density was associated with higher PM2.5 [16]. However, no studies have simultaneously and holistically considered how built environment configurations or profiles (that encompass multiple aspects acting together) are related to other city features across the universe of Latin American cities.

One empirical approach to managing the complexity of describing widely varying cities, and the interconnectedness of built environment attributes that characterize them, is to develop multi-dimensional typologies or profiles. Profiles are ways of describing units according to an ensemble of characteristics and allow for further studies into the mechanisms that connect the built environment to health. Profiles serve both as descriptions of current conditions and as triggers for policy action, either to address concerns or to transform the built environment in pursuit of specific planning or policy goals.

There are several existing examples of profiles based on indicators involving the built environment. These profiles include 1) transit-oriented development profiles [30], 2) bus rapid transit-oriented development profiles [31], 3) environmental profiles based on indicators of air pollution and biomass burning [32], 4) global city types based on visual classification of urban design and land transport [33], 5) urban form profiles of large metropolitan areas [7], and 6) typologies for urban edge dimensions at different scales [34]. To date, no studies have developed profiles of urban areas in Latin America, and none have focused on their association with social determinants of health and air pollution.

This study is innovative and makes unique contributions to the current literature evaluating relationships of built environment with social and environmental determinants of health. First, this study is part of the SALURBAL project [35] with the key aim of quantifying the contributions of city-level built environment factors to differences in health and health inequality among and within cities. The SALURBAL project is unique due to the inclusion of a large number of small, medium, and large cities from Latin America and the harmonization of data from multiple sources. These sources include satellite imagery of urban form, street design indicators from OpenStreetMaps, satellite-derived PM measures, and socioeconomic indicators from national censuses [36]. Second, we take into account the interconnectedness of urban form indicators by identifying profiles using finite mixture modeling (FMM). FMM is a useful approach to uncover latent classes to estimate the conditional probabilities of the observations to each class. Finally, we assess the relationship between social and environmental determinants of health with the built environment across a large scale of cities in Latin America.

The aims of our study were to: 1) empirically identify profiles of the urban landscape and street morphology of 370 cities of 100,000 residents of more across the region and 2) to evaluate the associations of these urban landscapes and street design profiles with social determinants of health (education, poverty, sewage connection, congestion) and air pollution. Characterizing the built environment profiles of cities is critical to understanding the impact of built environment features on health and environmental outcomes and to the development of urban policies that promote health and environmental sustainability. These types of analyses may inform global and local policies linked to the New Urban Agenda and the Sustainable Development Goals (SDGs), particularly SDG 11– Make cities and human settlements inclusive, safe, resilient and sustainable and its interaction with SDG 3-of health and wellbeing [37].

## Methods

### SALURBAL cities

The SALURBAL project includes 11 countries in Latin America: Argentina, Brazil, Chile, Colombia, Costa Rica, El Salvador, Guatemala, Mexico, Nicaragua, Panamá, and Perú. Additional details on the multilevel data structure of the study have been published earlier [36]. Briefly, to identify eligible cities ($\geq$100,000 inhabitants as of 2010), SALURBAL cities were selected using two databases (the Atlas of Urban Expansion (AUE) and citypopulation.de (CP) from official census data). Boundaries were based on official administrative boundaries of sub-city units in each country that encompassed the visual urban extent of each city. We obtained administrative boundaries from official country sources and overlayed those onto satellite imagery of each city to determine the urban extent [36]. The lists of the databases were matched, and differences were resolved using satellite imagery and population estimates. These 370 units are henceforth referred to as "cities" (or Level 1 administrative units). We chose this geographic definition because it matches the area used in mortality and health surveys of SALURBAL while accounting for the urban extent.

### SALURBAL built environment domains

Given their emerging importance [7, 38], and the possibility of measuring them reliably across cities and countries, we focused on two built environment domains: the urban landscape and street design. These domains were analyzed separately to account for potential associations between different profiles within each domain and to allow separate investigation of their association with socioeconomic, environmental or health outcomes.

## Urban landscape domain

The urban landscape domain measures the configuration of urban development within each city. We selected six urban landscape metrics that represent the subdomains of fragmentation, isolation, and shape of developed urban areas to measure the degree of development (dis)connectedness, the proximity/isolation of development, and the complexity of the overall shape. We also included urban area and population size (Table 1). The three landscape subdomains, in addition to total urban area and population, encompass attributes that can be used to characterize urban development. Previous studies have developed various strategies to measure urban form [10, 39]. Furthermore, a number of spatial metrics have been proposed to assess landscape configuration based on geometry and spatial relationships among discrete units, or patches [40–42]. To more effectively identify typologies of urban morphology landscape structure for SALURBAL cities, in this study, we chose a parsimonious set of landscape metrics based on the principles of strength, universality, and consistency [40]—i.e., metrics that represent different components of landscape structure such as size, shape, texture, and contiguity [10, 43].

The total urban area represents the overall spatial extent, which may reflect the stage of urban development, population size, and regional importance of cities as economic or cultural

**Table 1. Urban development landscape and street design domains, subdomains, and population metrics.**

| Subdomains | Metric | Abbreviation | Formula | Description |
|---|---|---|---|---|
| Area | Total urban area | TUA | $TUA_i = \sum_{j \in i, j \in urban} A_j$ | $A_j$ refers to the area of 30m x 30m gridcell j within the geographic unit i and categorized as urban. |
| Fragmentation | Number of Urban Patches (N) | NUP | $NUP_i$ | Where $NUP_i$ refers to the number of urban patches in city i |
| | Patch Density (N/km2) | PD | $PD_i = \frac{NUP_i}{TUA_i}$ | Where $NUP_i$ refers to the number of urban patches in city i and $TUA_i$ refers to the total urban area in city i. |
| | Area-weighted Mean Patch Size (km2/N) | AWMPS | $AWMPS_i = \frac{\sum_{j \in i} \frac{UA_j^2}{TUA_i}}{NUP_i}$ | Where $UA_j$ refers to the area of urban patch j located in the city i. |
| | Effective Mesh Size (km2) | EMS | $EMS_i = \frac{\sum_{k \in i} UA_k^2}{TUA_i}$ | Where $UA_j$ refers to the area of urban patch j located in the city i. |
| Shape | Area-weighted Mean Shape Index | AWMSI | $AWMSI_i = \frac{\sum_{k \in i} \frac{SHPINDX_k * UA_k}{TUA_i}}{NUP_i}$ | Where $SHPINDX_k$ refers to the shape index of urban patch $k$ inside city $i$, specifically, shape index is the ratio of the actual perimeter of a patch to the minimum perimeter possible for a maximally compact patch with the same size. $UA_k$ refers to the area of urban patch $k$ located in the city $i$. |
| Isolation | Area-weighted Mean Nearest Neighbor Distance (meters) | AWMNND | $AWMNND_i = \frac{\sum_{k \in i} \frac{NNHG_k * UA_k}{TUA_i}}{NUP_i}$ | Where $NNGH_k$ is the nearest neighbor distance of urban patch k in city i, $UA_k$ refers to the area of urban patch k located in the city i |
| Street connectivity | Street density (m/km2) | SD | $SD_i = \frac{\sum_{j,k \in E_i} Length_{j,k}}{area_i}$ | Where $area_i$ is the area in km$^2$ of city i and length(j,k) is the length in km of edge (j,k) in the set $E_i$ of all the edges in the street network of city i. |
| | Intersection density (nodes/km2) | ID | $ID_i = \frac{|N_i|}{area_i}$ | Where $area_i$ is the area in km$^2$ of city i and $|N_i|$ is the amount of nodes in the street network. |
| | Streets per node average (streets/nodes) | SNA | $SNA_i = \frac{\sum_{j \in N_i} \delta_j}{|N_i|}$ | Where $N_i$ is the set of nodes in the underlying street network of city i, $|N_i|$ is the size of $N_i$, and $\delta_j$ is the amount of edges connecting node j to other nodes in the network. |
| Street length | Street length average (meters) | SLA | $SLA_i = \frac{\sum_{j,k \in E_i} Length_{j,k}}{|E_i|}$ | Where $E_i$ is the set of edges in the underlying street network of city i, $|E_i|$ is the size of $E_i$, and length(j,k) is the length in km of edge (j,k) in the set $E_i$. |
| Directness | Circuity average | CA | $CA_i = \frac{1}{n} \sum_{j,k \in E_i} \frac{length_{j,k}}{STdistance_{j,k}}$ | Where Ei is the set of edges in the underlying street network of city i, length (j,k) is the length in km of edge (j,k) in the set Ei, STdistance(j,k) is the straight length distance in km of edge (j,k) in the set E |
| Population | Population | Population | NA | NA |

centers, which may to some extent determine their morphology based on differences in planning based on size.

Fragmentation metrics (Table 1) generally indicate the absolute and relative quantity of patches on the landscape and the degree to which urban form is not contiguous [44, 45]. A fragmented urban landscape has interstitial non-urban spaces, whereas a non-fragmented urban landscape is continuously developed. To characterize fragmentation, we used the following metrics: the number of urban patches, patch density, area-weighted mean patch size, and effective mesh size. An urban patch was defined as a contiguous area of urban development. Number of urban patches (NUP) presents the total count of disconnected units, while patch density (PD) provides a relative quantity of urban patches. Both greater NUP and greater PD indicate a greater degree of fragmentation. Area-weighted mean patch size (AWMPS) represents a weighted mean of individual patch areas, where larger patches have more influence on the metric value. This allows prioritizing larger patches containing more contiguous urban areas in the calculation and reducing the effect of small-sized development pockets. Higher value of AWMPS indicates lower fragmentation (i.e., prevalence of larger patches). Finally, we also used effective mesh size (EMS) which by design (Table 1) represents the expected size of a spatial region accessible for movement from a randomly chosen starting point without encountering a barrier, i.e., patch edge [46]. EMS is related to the probability that two points selected randomly within the study region are connected, i.e., fall within the same urban patch [47] with greater values indicating lower fragmentation. EMS provides a useful basis for comparing fragmentation among regions of different sizes and different proportions of developed areas within their boundaries [46, 47].

Shape is a measure of development compactness and complexity [44]. We used the area-weighted mean shape index to assess shape. The shape index is a ratio of the actual perimeter of a patch to the minimum perimeter possible for a maximally compact patch of the same size. The shape index ranges from 1 to infinity. An index of 1 represents a maximally compact patch, while larger values reflect shapes that are less compact and more complex [44]. Complexity of the shape of urban border is an important aspect of their form reflecting the characteristics of sprawl, geographic constraints on development, and the potential spatial interactions between the city and surrounding landscape [10, 41, 48, 49]. Area-weighted mean shape index (AWMSI) measures the complexity of the urban patch perimeter relative to the ideal compact shape with the same size, averaged among urban patches with higher influence of larger patches (Table 1). This measure provides a useful basis for distinguishing among geometric forms of urban sprawl with different compactness or different geographic constraints affecting the overall shape.

Area-weighted mean nearest neighbor distance (AWMNDD, Table 1) characterizes patch isolation as the average minimal edge-to-edge distance between discrete patches, weighted by patch area so that distances involving larger patches have a stronger influence on the metric value. While mean nearest-neighbor distance is commonly used in urban studies [48, 50], using its area-weighted form allows accounting for potential differences in movement or other distance-sensitive effects between urban patches based on contrasts in their population, amenities, and similar factors.

To create the metrics, we used the 2012 urban footprint data (in 30m x 30m grid cells) from the Global Urban Footprint project [51]. Metrics were calculated based on 30m x 30m grid cells using the FRAGSTATS 4.2 software package [42].

**Population.**   Population size measures the total number of residents within the administrative boundary, as reported in Worldpop for the year 2015 adjusting for United Nations' country-level population projections [52].

## Street design domain

Within the street design domain, we selected five metrics that represent the subdomains of street connectivity, street length, and directness (Table 1). We selected these metrics as they capture the cities' general street network structure and because they have been used in multiple studies allowing comparability [10, 38, 53]. To calculate these metrics, we used each geographic unit extracted from the street network of OpenStreetMap in 2017–2019. These metrics were calculated using the OSMNx software package [53, 54]. OpenStreetMap (OSM) data was approximately 83% complete by 2017.

To assess street connectivity, we used the following metrics: intersection density, street density, and the street per node average (Table 1). Intersection density is the number of intersections per km$^2$. Street density is the length of streets per km$^2$. Street per node average measures the mean number of streets meeting at each intersection of the street network. The higher the value of these metrics, the greater the degree of connectivity, thus enabling more direct travel between two points using existing streets [53].

Street length is a measure of the total average length of the street network between intersections. Small values represent a shorter average street network and larger values represent a longer average street length network [53].

Directness is a measure of the average ratio of network distances to straight-line distances from every node in the street network to every other node. We used circuity to assess directness [53]. This indicator ranges from 1 to infinity, where a value of 1 denotes no circuity (highest directness), and higher values denote higher circuity (lower directness).

## Social determinants of health and air pollution outcome variables

**Socio-economic metrics.** To assess socioeconomic conditions of households within cities, we included three metrics: the proportion of households with piped water access inside the dwelling (water in dwelling), the proportion of households with more than three people per room (overcrowding per room), and the proportion of the population aged 25 or older who completed primary education or above (completed primary or higher). Specifically, 25 years is the minimum age standard for measuring adult educational attainment defined by UNESCO and OECD assuming that most adults will have already completed basic, high school and college education in most countries [55]. These metrics were standardized among countries to be adequately comparable. Socioeconomic indicators were obtained from the national census bureau's [36].

**Traffic congestion.** To measure the average traffic delay due to congestion, we used the urban travel delay index (UTDI) [56]. The UTDI is a proxy measurement for congestion in the street network calculated as the percentage difference between the travel time during peak-hour traffic and the free-flow travel time. For each city, we calculated the travel times for 30 random origin-destination point pairs in the street network during seven-time points in the AM peak of a typical weekday and the midnight hours in June 2019. The Google Maps API was used for these calculations [57].

**PM2.5 annual mean concentration (ug/m$^3$).** To measure air pollution concentration, we used the annual mean concentration of PM2.5 (ug/m$^3$) estimated from satellite measurements obtained from the Atmospheric Composition Analysis Group of the Dalhousie University Annual means of 2016 gridded format with each grid cell representing 0.01 degrees by 0.01 degrees (~ 1.1km by 1.1km) [36]. The mean of all cells in the city was used to characterize air pollution for the city. Cells that were only partly included in the city were apportioned based on the area of overlap. For our aim of determining associations between different built environment city profiles and differences in city-level air pollution concentrations, the most

accurate approach is to use an unweighted air pollution metric that spatially matches the urban landscape and street design characteristics.

## Profiles identification

**Model-based finite-mixture modeling.** FMM was used to identify the profiles of the 370 cities according to the landscape and street design domains. FMM is a family of statistical models used to identify latent constructs, allowing for the identification of city profiles (or clusters of cities) to summarize the multidimensional nature of the city-level metrics of urban landscape and street design [58]. FMM uses a mixture of distributions to model the heterogeneity of group structures. This methodology is defined in terms of the measurements and their probability density functions. Let $Y_1,...,Y_n$ the realizations of the $p$ measurements in the sample $Y$ [58]. The probability density function $f(y_i)$ can be described as:

$$f(y_j) = \sum_{i=1}^{g} \pi_i * f_i(y_j) \tag{1}$$

where $f_i(y_j)$ denotes the component densities of the mixture, $g$ is the number of components of the finite mixture distribution, $\pi_i$ are called the weights or conditional probabilities to belong to each component. These weights are nonnegative quantities that sum one:

$$0 \leq \pi_i \leq 1 \quad (i = 1, \ldots, g)$$
$$\sum_{i=1}^{g} \pi_i = 1 \tag{2}$$

The value of $g$ and the conditional probabilities $\pi_i$ are unknown and must be inferred from the data [58].

We used a formative approach to conduct the FMM under the following theoretical statements: 1) definition of the nature of the latent classes (preexistence of latent classes), 2) causality from metrics to classes (direction of causality between metrics and latent class), and 3) metrics define the class (characteristics of metrics used to define the latent class) [59]. First, correlation among urban landscape indicators, street design, social determinants of health and air pollution were evaluated using a correlation matrix and the Pearson correlation coefficient. Second, we fitted a model for each domain (urban landscape and street design) specifying a different number of latent profiles to be extracted. Third, we assessed bivariate correlations between indicators of each domain. To select the number of latent profiles, we examined goodness-of-fit using the Bayesian Information Criteria (BIC) and classification using entropy [60]. We used the elbow method with both indicators, using BIC as the stop criteria and assessing whether classes were meaningfully different using entropy, to improve the interpretability of the results. Fourth, we assigned cities to profiles based on their most likely profile membership (modal assignment) and used descriptive statistics to assess whether the profiles captured meaningful similarities and differences across cities. Finally, we assigned the labels and icons to each profile based on the percentile distribution of each indicator within the profiles. We used three category labels: 'low' when the 50th percentile of the indicator of interest in profile $i$ is lower than the 25th percentile for the overall profiles indicators [$P_{i50} < P_{25}$], 'moderate' when the 50th percentile of the indicator of interest in profile $i$ is between the 25th and the 75th percentile for the overall profiles indicators [$P_{25} \leq P_{i50} \leq P_{75}$], and 'high' when the 50th percentile of the indicator in profile $i$ is higher than the 75th percentile for the overall profiles indicators [$P_{i50} > P_{75}$]. The FMM models were fitted in R [61], using the FlexMix package [62] and the expectation maximization (EM) algorithm for standard error estimation.

To study associations of profiles with social determinants of health, and air pollution we fitted two-level linear mixed models (cities nested within countries) using the lme4 package [63], with a random intercept for each country. The equation for the first level was:

$$Y_{ij} = b_{0j} + b_{1j}X_{ij} + \varepsilon_{ij}, \ \varepsilon_{ij} \sim N(0, \sigma^2) \qquad (3)$$

where $Y_{ij}$ is the outcome for the $i^{th}$ individual in the $j^{th}$ group, $b_{0j}$ is the intercept for the $j^{th}$ group, $b_{1j}$ is the coefficient for the matrix of variables for the $j^{th}$ group, $X_{ij}$ is the variables matrix for the $i^{th}$ individual in the $j^{th}$ group and $\varepsilon_{ij}$ is the $i^{th}$ individual error in the $j^{th}$ group. The outcome $Y_{ij}$ correspond to the social determinants of health and air pollution. The variables used to define the $X_{ij}$ matrix were the urban landscape and street design profiles.

The equation for the second level was:

$$b_{0j} = \gamma_{00} + \gamma_{01}C_j + U_{0j} \quad U_{0j} \sim N(0, \tau_{00})$$
$$b_{1j} = \gamma_{10} + \gamma_{11}C_j + U_{1j} \quad U_{1j} \sim N(0, \tau_{11}) \qquad (4)$$
$$cov(U_{0j}, U_{1j}) = \tau_{01}$$

where $\gamma_{00}$ is the common intercept across groups, $\gamma_{10}$ is the common slope associated with $X_{ij}$ across groups, $\gamma_{01}$ is the change in the intercept per unit change $C_j$, $\gamma_{11}$ is the change in the slope per unit change $C_j$, $U_{0j}$ is the unique deviation of the intercept of each group from the overall intercept, $U_{1j}$ is the unique deviation of the slope within each group from the overall slope, $\tau_{00}$ is the variance of the group specific intercept and $\tau_{11}$ is the variance of the group specific slope. All models were adjusted by area and population (both $\log_{10}$ transformed).

**Visualization of profiles.** To support the exploration of the profiles, we created a visual analytics tool. The tool links geospatial data visualization units of profiles to descriptive statistics of indicators and profiles. We used open-source visualization libraries such as D3.js [64] and Vega-Lite [65] in JavaScript for running in the browser (https://salurbal.github.io/profiles/).

## Results

### Urban landscape indicators

Overall, urban landscape indicators varied between countries and cities (S1 Table; visualization tool). Within the subdomain of fragmentation, the number of urban patches (p50 [p25; p75]) (401.5 [250.5;740.0]) and patch density (0.29 [0.12;0.56]) varied significantly. The most fragmented country characterized by the highest mean number of urban patches and high patch density was Costa Rica (2,888 urban patches; 0.93 urban patches/km$^2$), and the country with lowest patches and patch density was Perú (173 urban patches; 0.14 urban patches/km$^2$). The city with the highest number of urban patches (7,549 urban patches) was Buenos Aires (Argentina) and the city with the highest patch density (2.50 urban patches/km$^2$) was Quetzaltenango (Guatemala). In contrast, the city with the lowest number of urban patches (31 urban patches) was Puno (Perú) and the lowest patch density (0.008 urban patches/km$^2$) was Antofagasta (Chile). Furthermore, the smallest area-weighted mean patch size showed a large variability (1880.2 [1082.4;3928.5]). The country with the lowest area-weighted mean patch size and effective mesh size, indicating small patches, was Nicaragua (734.7 km$^2$/urban patches; 17.5 km$^2$) while Costa Rica had the largest patches (17494.5 km$^2$/urban patches; 1941.2 km$^2$). The most fragmented city due to the lowest area-weighted mean patch size and effective mesh size (35.02 km$^2$/urban patches; 0.65 km$^2$) was Petropolis (Brazil). The city with the highest area-weighted mean patch size (132006.47 km$^2$/urban patches) was Buenos Aires (Argentina) and the highest effective mesh size (41125.55 km$^2$) was Sao Paulo (Brazil).

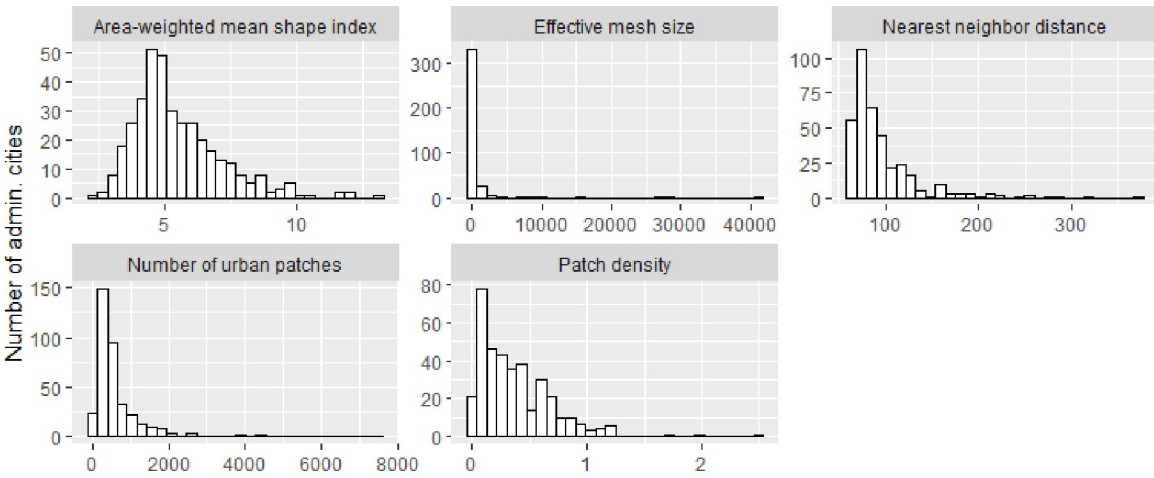

**Fig 1. Histograms of the distribution of urban landscape metrics.**

Within the subdomain of shape, the mean area-weighted mean shape index showed a large variability (4.98 [4.33;6.27]). The country with the highest mean area-weighted mean shape index, indicating a complex shape, was Costa Rica (12.01), and the country with the most compact shape was Nicaragua (3.99). The most complex city (13.26) was Buenos Aires (Argentina) and the most compact city (2.40) was Rio Gallegos (Argentina).

Within the subdomain of isolation, the highest mean area-weighted mean nearest neighbor distance showed a large variability (82.76 [72.28;102.83]). The country with the highest mean area-weighted mean nearest neighbor distance, indicating higher isolation, was Mexico (91.7 meters), and the less isolated country was Costa Rica (65.4 meters). The city with the highest isolation (370.13 meters) was Quibdó (Colombia) and the city with the lowest isolation (62.48 meters) was Rio de Janeiro (Brazil). Histograms of each Urban landscape metric can be found in Fig 1.

## Street design domain

Overall, street design indicators varied between countries and cities (S1 Table; visualization tool). Within the subdomain of street connectivity, the street density (1137.4 [529.3;1956.9]) and intersection density (4.63 [2.02;9.07]) showed large variability. The country with the highest street and intersection densities was Guatemala (4075.9 m/km$^2$; 20.42 intersections/km$^2$), and the country with the lowest street connectivity was Argentina (591.4 meters/km2; 1.73 intersections/km$^2$). The city with the highest intersection density (42.08 intersections/km$^2$) was Santiago (Chile) and the city with the highest street density (6742.7 m/km$^2$) was Sao Paulo (Brazil). The city with the lowest intersection density (0.11 intersections/km$^2$) and street density (25.3 m/km$^2$) was Quibdó (Colombia).

The indicator of street length average (134.7 [118.6 163.8]) showed a large variability. The country with the highest street length average was Argentina (164.9 m), and the country with the shortest streets was Guatemala (113.1 m). The city with the highest average street length (510.1 m) was Barreiras (Brazil) and the city with the shortest average street length (82.8 m) was Santiago (Chile).

Within the subdomain of directness, the circuity average (1.0646 [1.0470 1.0891]) showed a large variability. The country with the highest average circuity, indicating less direct streets was Costa Rica (1.12), and the country with the most direct streets was Perú (1.05). The city

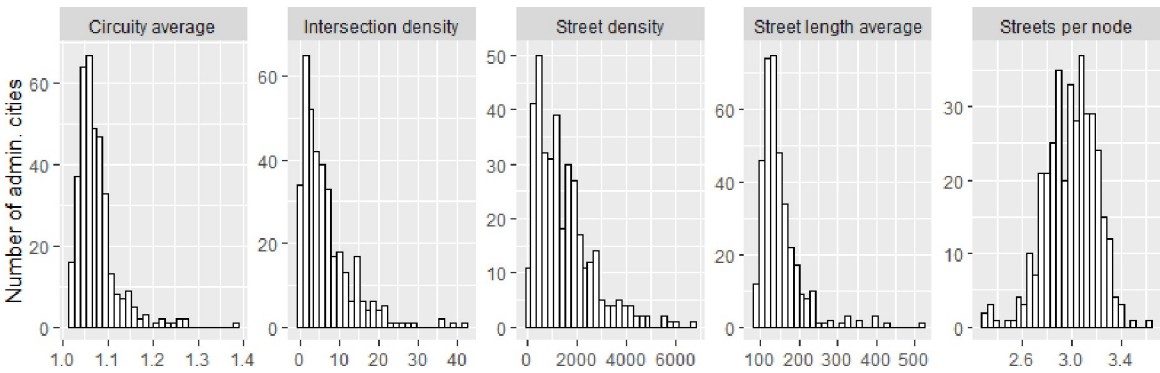

**Fig 2. Histograms of the distribution of street design metrics.**

with the least direct streets (1.37) was Huaraz (Peru) and the city with the most direct streets (1.01) was Santa Rosa (Argentina). Histograms of each Street design metric can be found in Fig 2.

## Correlation among urban landscape and street design domains

We found that the largest positive Pearson correlation coefficient among metrics of urban landscape and street design domains was between street density and patch density (0.75, Table 2), indicating a positive correlation between high street connectivity and high fragmentation. The largest negative Pearson coefficient was between street density and area-weighted mean nearest neighbor distance (-0.44), indicating a negative correlation between low street connectivity and high isolation. Furthermore, in the subdomain of fragmentation, the number of urban patches and patch density were positively correlated with area-weighted mean patch size and effective mesh size. These findings indicate the co-existence of high fragmentation due to high patch density and the dominance of one or more large patches that increase mean patch sizes in some cities. One such example is San Jose (Costa Rica), characterized by high fragmentation due to its high patch density, but with large mean patch size due to a large patch of continuous development.

## Profiles

**Urban landscape profiles.** The FMM yielded four profiles according to the BIC criteria (Table 3). The entropy criterion for dissolution was 0.92 indicating a high certainty in the classification (S1 and S2 Figs). The first profile contained 168 cities (45.4%) and was characterized by cities with moderate patch density ($[P_{25}; P_{75}]$, [0.12; 0.56]), moderate area-weighted mean patch size (1082.4; 3928.5), irregular shape (4.33; 6.27) and moderate isolation (72.2; 102.8). We labeled this profile *proximate stones*. Cities with a high conditional probability of belonging to this profile included Pocos de Caldas, Araras, Birigui and Parnaiba (Brazil) and San Nicolas de los Arroyos (Argentina). The second profile was characterized by cities with lower patch density ($[<P_{25}]$, [<0.12]), lower area-weighted mean patch size (<1082.4), a compact shape (<4.33) and higher isolation ($[>P_{75}]$, [>102.8]). This profile was labeled *scattered pixels* and included 91 cities (24.6%). Cities with a high conditional probability of belonging to this profile included Barreiras (Brazil), Quibdó (Colombia), and Chilpancingo, Fresnillo and Tecoman (Mexico). The third profile was characterized by cities with a moderate patch density (0.12; 0.56), larger area-weighted mean patch size (>3928.5), complex shape (>6.27) and moderate isolation (72.2;102.8). This profile was labeled as *proximate inkblots* and included 90 cities

**Table 2. Pearson correlation coefficients among urban landscape, street design domains, population metrics, social determinants of health and air pollution.**

| | | Fragmentation | | | | Shape | Isolation | Street connectivity | | | Street length | Directness | Population metrics | | Social determinants of health | | | | Air pollution |
|---|---|---|---|---|---|---|---|---|---|---|---|---|---|---|---|---|---|---|---|
| | | NUP | PD | EFS | AWMPS | AWMSI | AWMENND | ID | SD | SPNA | SLA | CA | TUA | POP | PHWPW | PHWM3P | PPA25WCP | UTDI | PM2.5 AMC |
| Fragmentation | NUP | 1,00 | | | | | | | | | | | | | | | | | |
| | PD | 0,39 | 1,00 | | | | | | | | | | | | | | | | |
| | EFS | 0,64 | 0,20 | 1,00 | | | | | | | | | | | | | | | |
| | AWMPS | 0,73 | 0,20 | 0,96 | 1,00 | | | | | | | | | | | | | | |
| Shape | AWMSI | 0,63 | 0,37 | 0,49 | 0,60 | 1,00 | | | | | | | | | | | | | |
| Isolation | AWMENND | -0,18 | -0,39 | -0,14 | -0,18 | -0,36 | 1,00 | | | | | | | | | | | | |
| Street connectivity | ID | 0,42 | 0,65 | 0,52 | 0,53 | 0,53 | -0,41 | 1,00 | | | | | | | | | | | |
| | SD | 0,47 | 0,75 | 0,50 | 0,50 | 0,55 | -0,44 | 0,97 | 1,00 | | | | | | | | | | |
| | SPNA | -0,09 | -0,40 | -0,01 | 0,03 | -0,11 | 0,07 | -0,13 | -0,21 | 1,00 | | | | | | | | | |
| Street length | SLA | -0,08 | -0,25 | -0,13 | -0,17 | -0,28 | 0,43 | -0,44 | -0,36 | 0,02 | 1,00 | | | | | | | | |
| Directness | CA | -0,14 | -0,06 | -0,08 | -0,14 | -0,13 | 0,10 | -0,17 | -0,11 | -0,50 | 0,29 | 1,00 | | | | | | | |
| Population metrics | TUA | 0,88 | 0,29 | 0,90 | 0,95 | 0,64 | -0,20 | 0,55 | 0,55 | -0,02 | -0,17 | -0,15 | 1,00 | | | | | | |
| | POP | 0,76 | 0,26 | 0,93 | 0,94 | 0,59 | -0,18 | 0,59 | 0,56 | -0,05 | -0,18 | -0,09 | 0,95 | 1,00 | | | | | |
| Social determinants of health | PHWPW | 0,15 | 0,13 | 0,06 | 0,08 | 0,10 | -0,12 | 0,11 | 0,16 | 0,07 | 0,02 | 0,00 | 0,10 | 0,06 | 1,00 | | | | |
| | PHWM3P | -0,11 | -0,07 | -0,01 | -0,03 | -0,10 | 0,15 | -0,09 | -0,14 | -0,02 | -0,01 | 0,00 | -0,05 | -0,01 | -0,68 | 1,00 | | | |
| | PPA25WCP | 0,19 | -0,06 | 0,11 | 0,18 | 0,18 | -0,13 | 0,09 | 0,03 | 0,05 | -0,12 | -0,09 | 0,19 | 0,17 | 0,14 | 0,13 | 1,00 | | |
| | UTDI | 0,10 | -0,05 | 0,09 | 0,11 | 0,09 | 0,02 | 0,04 | 0,03 | 0,13 | 0,03 | -0,09 | 0,11 | 0,10 | 0,00 | -0,01 | 0,06 | 1,00 | |
| Air pollution | PM2.5 AMC | 0,14 | 0,34 | 0,23 | 0,20 | 0,17 | -0,20 | 0,31 | 0,33 | -0,09 | -0,15 | -0,16 | 0,21 | 0,21 | 0,03 | 0,11 | 0,14 | -0,08 | 1,00 |

NUP: Number of urban patches.

PD: Patch density.

EFS: Effective mesh size.

AWMPS: Area weighted mean patch size.

AWMSI: Area weighted mean shape index.

AWMENND: Area weighted mean euclidean nearest neighbor distance.

ID: Intersection density.

SD: Street density.

SPNA: Streets per node average.

SLA: Street length average.

CA: Circuity average.

TUA: Total urban area.

POP: Population.

PHWPW: Proportion of households with piped water access.

PHWM3P: Proportion of households with more than 3 people per bedroom.

PPA25WCP: Proportion of the population aged 25 or older who completed primary or above.

UTDI: Urban Travel Delay Index.

PM2.5 AMC: PM2.5 annual mean concentration (ug/m3).

**Table 3. Urban landscape and street design profiles.**

| Urban Landscape | |
|---|---|
| Label | Description |
| Proximate stones | Cities with moderate patch density and moderate area weighted mean patch size, patches with irregular shape and moderate isolation. |
| Scattered pixels | Cities with lower patch density and lower area weighted mean patch size, patches with compact shape and higher isolation. |
| Proximate inkblots | Cities with moderate patch density and higher area weighted mean patch size, patches with complex shape and moderate isolation. |
| Contiguous large inkblots | Cities with higher patch density and higher area weighted mean patch size, patches with complex shape and lower isolation. |
| **Street design** | |
| Label | Description |
| Semi-hyperbolic grid | Cities with moderate street connectivity, streets with moderate length and moderate directness streets. |
| Labyrinthine | Cities with low street connectivity, streets with moderate length and moderate directness streets. |
| Spiderweb | Cities with higher street connectivity, shorter streets, and moderate directness streets. |
| Hyperbolic grid | Cities with moderate street connectivity, larger streets, and lower directness. |

The illustrations for urban landscape and street design metrics are gridded urban raster datasets developed in SALURBAL analyses of urban footprint.

*(*24.3%). Cities with a high conditional probability of belonging to this profile included Cartagena (Colombia), and Chihuahua, Hermosillo, Mexicali and Torreon (Mexico). The fourth profile was characterized by cities with a higher patch density (>0.56), higher area-weighted mean patch size (>3928.5), complex shape (>6.27) and lower isolation (<72.2). This profile was labeled as *contiguous large inkblots* and included 21 cities (5.7%). Cities with a high conditional probability of belonging to this profile included Buenos Aires (Argentina), and Belo Horizonte, Curitiba, Porto Alegre y Recife (Brazil). For specific information of profiles by country and cities see visualization tool (S2 Table).

**Street design profiles.** The FMM yielded four profiles according to the BIC criteria (Table 3). The entropy criterion for dissolution was 0.88 indicating a high certainty in the classification (S1 and S2 Figs). The first profile was characterized by cities with moderate street connectivity due to moderate intersection density ($[P_{25}; P_{75}]$, [2.01; 9.07]) and street density (529.3; 1956.9), streets with moderate length (118.6; 163.7) and moderately direct streets (1.0470; 1.0891). This profile was labeled *semi-hyperbolic grid* and included 130 cities (35.1%). Cities with a high conditional probability of belonging to this profile included Sao Carlos, Araras, Londrina, Marilia and Arapongas (Brazil). The second profile was characterized by cities with low street connectivity due to low intersection density ($[<P_{25}]$, [<2.01]) and street density (<529.3), streets with moderate length (118.6; 163.7) and streets with moderate directness (1.0470; 1.0891). This profile was labeled *labyrinthine* and included 110 cities (29.7%). Cities with a high conditional probability of belonging to this profile included Rio Cuarto, Rio Gallegos and Villa Mercedes (Argentina), and Uruguaiana and Barreiras (Brazil). The third profile was characterized by cities with higher street connectivity due to high intersection density ($[>P_{75}]$, [>9.07]) and street density (>1956.9), shorter streets (<118.6) and streets moderately directed (1.0470; 1.0891). This profile was labeled as *the spiderweb* and included 80 cities (21.6%). Cities with a high conditional probability of belonging to this profile included Buenos Aires (Argentina), Quetzaltenango (Guatemala), and Recife, Rio de Janeiro and Sao Paulo (Brazil). The fourth profile was characterized by cities with moderate street connectivity due to

**Table 4. Overlap among urban landscape and street design profiles.**

| Street Design profiles | Urban Landscape profiles | | | |
|---|---|---|---|---|
| | Contiguous large inkblots | Proximate stones | Scattered pixels | Proximate inkblots |
| Labyrinthine | 0.00% | 12.10% | 53.40% | 5.30% |
| Semi-hyperbolic grid | 0.70% | 39.30% | 4.70% | 18.90% |
| Spiderweb | 24.70% | 8.40% | 0.00% | 31.80% |
| Hyperbolic grid | 0.00% | 19.10% | 8.50% | 2.90% |

moderate intersection (2.01; 9.07) and street density (529.3; 1956.9), long streets (>163.7) and streets with low directness (>1.0891). This profile was labeled *hyperbolic grid* and included 50 cities (13.5%). Cities with a high conditional probability of belonging to this profile included Petropolis, Teresopolis and Nova Friburgo (Brazil), Huaraz (Peru) and Manizales (Colombia). For specific information of profiles by country and cities see visualization tool (S2 Table).

**Overlap among urban landscape and street design profiles.** We found that the most frequent overlap between landscape and street design profiles was among *Labyrinthine* and *Scattered pixels* (53.44% of cities), characterizing cities with isolated patches and a less well-connected street design network. A substantive overlap between *Semi-hyperbolic grid* and the *Proximate stones* (39.26% of cities) profiles was also identified. The cities belonging to these two profiles were characterized by moderate connectivity and moderate fragmentation with irregular shape (Table 4).

**Urban landscape and street design profiles by city population and density.** Profiles differed by city population and population density (Table 5). Smaller and less dense cities were more likely to belong to the *proximate stones* and the *labyrinthine* profiles. In contrast, larger cities were more likely to belong to the *contiguous large inkblots* and the *spiderweb* profiles and whereas denser cities were more likely to belong to the *proximate stones* and the *semi-hyperbolic grid* profiles.

## Associations of urban landscape and street design profiles with social determinants of health and air pollution indicators

We found that profiles characterized by moderate to higher street connectivity and moderate to low directness (*semi-hyperbolic grid*, *spiderweb*, and *hyperbolic grid*) were positively associated with more piped water access and negatively associated with overcrowding. Cities with moderate patch density, moderate to larger patch size, irregular to complex shape and moderate isolation (*proximate stones* and *proximate inkblots*) were positively associated with higher congestion. The *semi-hyperbolic grid* and *spiderweb* profiles were positively associated with higher annual mean concentration of PM2.5 (Table 6).

## Visualization of profiles

The visualization tool (https://salurbal.github.io/profiles/) presents an interactive map illustrating the geographical distribution of the 370 cities. It includes a dashboard where users can select the profiles for urban landscape or street design, and then explore the distribution of cities within them.

## Discussion

Our study characterized the urban landscape and street design of 370 cities in Latin America and the association of these profiles with social and environment determinants of health We identified four profiles for the urban landscape and four for street design. The characteristics

**Table 5. Urban landscape and street design profiles classified by city population and density.**

| | Number of cities | Urban Landscape | | | | | | | |
| | | Proximate stones | | Scattered pixels | | Proximate inkblots | | Contiguous large inkblots | |
| | | N | Density | N | Density | N | Density | N | Density |
|---|---|---|---|---|---|---|---|---|---|
| Overall | 370 | 168 (45.4%) | 5925 [4833;8068] | 91 (24.6%) | 6971 [5084;10951] | 90 (24.3%) | 5873 [5029;7600] | 21 (5.7%) | 7274 [5963;9626] |
| [100k:250k) | 170 | 94 (55.3%) | 5249 [45378;7109] | 71 (41.8%) | 6278 [4977;10611] | 5 (2.9%) | 4636 [2720;6056] | 0 (0.0%) | NA |
| [250k:500k) | 95 | 60 (63.2%) | 6664 [5449;8268] | 18 (18.9%) | 9106 [6632;13159] | 17 (17.9%) | 5439 [4826;5989] | 0 (0.0%) | NA |
| [500k:1M) | 57 | 13 (22.8%) | 8136 [7480;11157] | 2 (3.5%) | 11095 [10062;12127] | 42 (73.7%) | 5621 [4872;6880] | 0 (0.0%) | NA |
| [1M:5M) | 41 | 1 (2.4%) | 18688 | 0 (0.0%) | NA | 26 (63.4%) | 7702 [6460;12840] | 14 (34.1%) | 6162 [5498;7478] |
| [5M:20M) | 7 | 0 (0.0%) | NA | 0 (0.0%) | NA | 0 (0.0%) | NA | 7 (100.0%) | 10710 [9515;13335] |
| P-value | | <0.001 | <0.001 | <0.001 | 0.346 | <0.001 | 0.026 | <0.001 | 0.018 |
| | Number of cities | Street Design | | | | | | | |
| | | Semi-hyperbolic grid | | Labyrinthine | | Spiderweb | | Hyperbolic grid | |
| | | N | Density | N | Density | N | Density | N | Density |
| Overall | 370 | 130 (35.1%) | 5850 [5067;7926] | 110 (29.7%) | 5937 [4580;7944] | 80 (21.6%) | 6942 [5391;9774] | 50 (13.5%) | 7662 [5279;11691] |
| [100k:250k) | 170 | 60 (35.3%) | 5480 [4839;8076] | 69 (40.6%) | 5710 [4521;7320] | 10 (5.9%) | 5132 [4422;6915] | 31 (18.2%) | 7333 [4771;10159] |
| [250k:500k) | 95 | 37 (38.9%) | 6629 [5286;8259] | 30 (31.6%) | 6730 [5248;9466] | 17 (17.9%) | 5893 [5222;7901] | 11 (11.6%) | 7436 [6264;21326] |
| [500k:1M) | 57 | 25 (43.9%) | 5832 [5085;7177] | 9 (15.8%) | 4810 [4438;8042] | 16 (28.1%) | 5935 [5432;8255] | 7 (12.3%) | 8309 [7544;13363] |
| [1M:5M) | 41 | 8 (19.5%) | 6276 [5582;6830] | 2 (4.9%) | 6302 [5480;7124] | 30 (73.2%) | 7566 [6015;11593] | 1 (2.4%) | 18688 |
| [5M:20M) | 7 | 0 (0.0%) | NA | 0 (0.0%) | NA | 7 (100.0%) | 10710 [9515;13335] | 0 (0.0%) | NA |
| P-value | | <0.001 | 0.554 | <0.001 | 0.722 | <0.001 | 0.012 | <0.001 | 0.204 |

The percentage of cities in each profile is provided in parenthesis.

The 95% confidence intervals are shown in squared brackets.

and composition of the profiles highlight the heterogeneity and complexity of cities in the region. When examining associations with social and environmental determinants of health, we did not find a specific profile that was consistently associated with lower air pollution and better social conditions. Profiles of cities including higher street connectivity with moderate directed streets exhibited better socioeconomic conditions, but more air pollution; while profiles including more fragmentation, less isolation and more complex shape were associated with more traffic congestion.

Prior applications of profiles have emphasized their descriptive nature and their use as triggers of policy change, either to address concerns such as fragmentation and isolation, or as an aspirational vision to be achieved. Yet, the profiles in this study were not consistently and unequivocally associated with positive environmental and social outcomes. These findings underscore the tradeoffs often present in areas of rapid growth and economic need: economic development and productivity improving housing and social development can have adverse

**Table 6. Multilevel modeling of the associations between social determinants of health and air pollution indicators with urban landscape and street design profiles.**

| Profiles | Urban Landscape (1) | Street Design (2) | (1)+(2) |
|---|---|---|---|
| **Proportion of households with piped water access (N = 370)** | | | |
| Total urban area (log10)* | **17.80 [9.35;26.25]** | **22.12 [14.09;30.15]** | **22.27 [13.59;30.94]** |
| Population (log10)* | **-14.57 [-22.49;-6.65]** | **-18.06 [-25.81;-10.32]** | **-17.60 [-25.61;-9.59]** |
| Scattered pixels | Referent | | referent |
| Contiguous large inkblots | 3.85 [-4.25;11.96] | | -1.89 [-10.70;6.91] |
| Proximate stones | **3.26 [0.56;5.95]** | | -0.01 [-3.24;3.20] |
| Proximate inkblots | **4.27 [0.10;8.44]** | | 0.01 [-4.83;4.86] |
| labyrinthine | | Referent | referent |
| Semi-hyperbolic grid | | **4.79 [2.29;7.28]** | **4.74 [1.72;7.75]** |
| Spiderweb | | **5.12 [1.77;8.47]** | **5.21 [1.31;9.11]** |
| Hyperbolic grid | | **6.21 [2.84;9.57]** | **6.22 [2.53;9.92]** |
| **Proportion of households with more than 3 people per bedroom (N = 365)** | | | |
| Total urban area (log10)* | **-3.70 [-6.07;-1.43]** | **-5.66 [-7.91;-3.42]** | **-5.08 [-7.50;-2.67]** |
| Population (log10)* | **3.87 [1.66;6.08]** | **5.07 [2.92;7.23]** | **4.87 [2.64;7.09]** |
| Scattered pixels | Referent | | referent |
| Contiguous large inkblots | **-2.46 [-4.71;-0.21]** | | -0.78 [-3.21;1.64] |
| Proximate stones | **-1.37 [-2.11;-0.62]** | | -0.53 [-1.42;0.35] |
| Proximate inkblots | **-2.04 [-3.19;-0.89]** | | -0.90 [-2.23;0.42] |
| labyrinthine | | Referent | referent |
| Semi-hyperbolic grid | | **-1.35 [-2.04;-0.67]** | **-1.04 [-1.86;-0.21]** |
| Spiderweb | | **-1.93 [-2.85;-1.01]** | **-1.57 [-2.64;-0.50]** |
| Hyperbolic grid | | **-2.24 [-3.16;-1.31]** | **-1.98 [-2.99;-0.96]** |
| **Proportion of the population aged 25 or older who completed primary of above (N = 370)** | | | |
| Total urban area (log10)* | **13.29 [9.10;17.47]** | **14.09 [10.00;18.17]** | **14.03 [9.67;18.40]** |
| Population (log10)* | **-5.11 [-9.02;-1.20]** | **-6.66 [-10.58;-2.73]** | **-5.65 [-9.66;-1.63]** |
| Scattered pixels | Referent | | referent |
| Contiguous large inkblots | -2.28 [-6.25;1.68] | | -3.18 [-7.57;1.19] |
| Proximate stones | 0.74 [-0.56;2.06] | | 0.28 [-1.31;1.88] |
| Proximate inkblots | 1.17 [-0.85;3.21] | | 0.55 [-1.85;2.95] |
| labyrinthine | | Referent | referent |
| Semi-hyperbolic grid | | 0.87 [-0.36;2.12] | 0.59 [-0.90;2.08] |
| Spiderweb | | 0.88 [-0.79;2.56] | 0.84 [-1.09;2.77] |
| Hyperbolic grid | | 1.19 [-0.49;2.87] | 1.06 [-0.76;2.89] |
| **Urban Travel Delay Index (N = 370)** | | | |
| Total urban area (log10)* | -4.49 [-9.96;0.98] | -2.28 [-7.69;3.13] | -5.25 [-10.97;0.46] |
| Population (log10)* | **14.69 [9.57;19.81]** | **13.30 [8.09;18.51]** | **15.10 [9.83;20.36]** |
| Scattered pixels | referent | | referent |
| Contiguous large inkblots | 1.66 [-3.54;6.86] | | 2.14 [-3.59;7.89] |
| Proximate stones | **2.71 [0.98;4.43]** | | **3.23 [1.13;5.32]** |
| Proximate inkblots | **4.40 [1.73;7.06]** | | **4.87 [1.72;8.02]** |
| labyrinthine | | referent | referent |
| Semi-hyperbolic grid | | 1.09 [-0.56;2.75] | -0.82 [-2.78;1.13] |
| Spiderweb | | 1.69 [-0.53;3.91] | -0.22 [-2.75;2.31] |
| Hyperbolic grid | | 0.28 [-1.95;2.51] | -1.27 [-3.67;1.13] |
| **PM2.5 annual mean concentration (ug/m3) (N = 370)** | | | |
| Total urban area (log10)* | **5.26 [2.21;8.30]** | **6.81 [3.89;9.73]** | **6.38 [3.23;9.52]** |

(*Continued*)

**Table 6.** (Continued)

| Profiles | | Urban Landscape (1) | Street Design (2) | (1)+(2) |
|---|---|---|---|---|
| | Population (log10)* | **-5.14 [-8.05;-2.23]** | **-5.95 [-8.81;-3.09]** | **-6.07 [-9.03;-3.11]** |
| | Scattered pixels | referent | | referent |
| | Contiguous large inkblots | **4.79 [1.60;7.98]** | | 2.25 [-1.22;5.72] |
| | Proximate stones | **1.80 [0.73;2.87]** | | 0.72 [-0.55;2.01] |
| | Proximate inkblots | **2.03 [0.38;3.67]** | | 0.27 [-1.64;2.20] |
| | labyrinthine | | referent | referent |
| | Semi-hyperbolic grid | | **1.84 [0.84;2.84]** | **1.57 [0.37;2.77]** |
| | Spiderweb | | **2.98 [1.65;4.31]** | **2.66 [1.11;4.21]** |
| | Hyperbolic grid | | **1.37 [0.03;2.71]** | 1.05 [-0.41;2.51] |

The 95% confidence intervals are shown in square brackets.

consequences for air pollution and congestion. Together, these profiles provide a novel conceptual perspective to understanding urban areas. The inclusion of cities with different sizes sheds new light on the organization of cities and can serve as indicators to guide decisions about urban planning, transport, and health.

Among the four profiles emerging in the urban landscape domain, the most common one was the *proximate stones* characterized by moderate fragmentation, moderate patch size, moderate isolation, and irregular shape. This profile contains 168 cities and is home to 16.3% of the population that resides in all cities included in this study. The least common profile was the *contiguous large inkblots*, characterized by the largest and highest number of patches, and including all cities with more than 5 million people (47.8% of the population in 7 cities). Similarly, four profiles emerged for the street design domain. The most common street design profile was the *semi-hyperbolic grid*, and home to 19.4% of the population living in 130 cities, characterizing cities with moderate connectivity, moderate street length, and moderate directness. The least common profile was the *hyperbolic grid* with 5.1% of the population living in 50 cities. This profile is characterized by a less directed street network.

The built environment profiles were associated with social determinants of health and air pollution. Street design profiles were associated with socioeconomic conditions. The *semi-hyperbolic grid*, *spiderweb*, and *hyperbolic grid* profiles were associated with higher access to piped water and less overcrowding, but not population education. The cities in these profiles exhibit higher socioeconomic conditions compared to the profile with less connectivity and moderate directness (*labyrinthine*). Likewise, the *semi-hyperbolic grid* and *spiderweb* profiles were associated with higher air pollution. Urban landscape profiles were associated with congestion. Specifically, the *proximate stones* and *proximate inkblots* profiles were associated with higher congestion.

The urban landscape profile characterizing the large cities, *contiguous large inkblots* characterized by the largest and highest number of patches is similar to what has been previously described in other multi-country studies that included only cities like Bogotá, Buenos Aires, Brasilia, Córdoba, Lima, and Santiago [7, 66]. These Latin American cities are denser and less complex in shape than cities in North America and Western Europe [7], but less dense than cities in sub-Saharan Africa and South-East Asia [25]. In fact, the density of Latin America is five times higher than the United States and three times higher than in Western Europe [26]. In addition, previous studies have shown the heterogeneity of the fragmentation and density of Latin American cities [10, 66]. Our study adds to previous studies by identifying new profiles for medium and small size cities that differ from the *contiguous large inkblots'* profile of

large capital cities. Specifically, these new profiles are characterized by smaller, more isolated, and less fragmented cities.

Additionally, when comparing metrics of fragmentation in our study with a study in the US that included 3097 counties, Latin American cities on average are more fragmented. The cities in our study have, on average, 2 times more urban patches and higher variability compared to US counties [45] (660 (SD:834) vs. 324(SD:355)) [45]. Compared to prior studies, our study has expanded the profiles of Latin American cities by including multiple cities with populations of one million or fewer inhabitants which are home to more than 100 million people (36.2%) and are mainly characterized by low to moderate patch density coexisting with moderate to high isolation. Future studies will need to longitudinally assess changes in these profiles to explain several environmental, economic, and development dynamics of the region.

In addition, previous studies in large and capital cities have shown that fragmentation and complexity are associated with more economic development [66], and higher income [7]. In contrast, a large study of Latin American cities authors found that cities with compact shape and high street density exhibit higher levels of productivity [10]. Our results revealed expanded typologies of fragmentation based on number of patches, patch size and isolation and showed different patterns in the relationship between fragmentation and economic development. Specifically, we found that the *scattered pixels* profile, which includes smaller patches plus more isolation, was associated with poorer socioeconomic conditions as reflected in water access and overcrowding. However, these associations were not significant when we adjusted for street design profiles, reflecting the strong association of street connectivity with water access and overcrowding. It has been hypothesized that street connectivity is associated with economies of agglomeration and city economic performance, in part, by facilitating better interaction between individuals and across people and firms [10]. Future studies should assess the associations of urban landscape and street design profiles with health outcomes and health inequalities.

Latin America cities have been characterized by having an urbanization without economic growth [25]. This region has been characterized as having, in part, high levels of urbanization with low economic development and high informality [25]. In terms of economic development, between 1913 to 2008, Latin American has had a significantly lower annual increase in their GDP compared to high income countries with similar levels of urbanization (81% in the USA and 73% in Europe) [25]. Future studies will need to investigate the association of urban landscape with measures of housing informality and accessibility (housing, jobs, education, health) to better understand the mechanisms by which urban landscapes and street design are associated with economic development and multiple socioeconomic indicators in Latin American cities.

The rapid urbanization in Latin America has also likely led to longer travel times and higher levels of congestion [67] with several of capital cities including Bogotá, Quito and Cali featured among the most congested in the world [68]. In fact, our results of the association between congestion with larger populations is consistent with previous work showing that congestion scales superlinearly with city population [69]. Furthermore, higher levels of fragmentation measured by patch density have been correlated with higher motorization rates that, in turn, could lead to more congestion costs [31, 66]. Our work further shows the association of congestion with profiles characterized by complex shapes, larger patches, and larger populations as compared to the profiles with *scattered pixels* profile. Current recommended interventions to decrease congestion include taxation on car use, car use restriction or congestion charge [15, 70]. Our study shows that urban form should also be considered when implementing and evaluating these interventions aimed at decreasing congestion.

Previous studies have shown that Latin American cities are characterized by high street and intersection density [10]. Our study adds to this previous literature by including additional metrics of directness and street length average and identifying street design profiles for large medium and small size cities. Specifically, these new profiles for medium and small cities are characterized as being less connected, with larger streets and moderate directed streets. These profiles overlapped mostly with the *proximate stones* and *scattered pixels* profiles of the urban landscape domain. Compared with the profile including the capital cities (*contiguous large inkblots*), these profiles are associated with high socioeconomic conditions (excluding the *labyrinthine profile*) and high levels of air pollution for the *semi-hyperbolic grid* profile.

Studies conducted in Latin America, the US and China assessing the relationship between air pollution and urban landscape metrics showed that higher fragmentation, higher population density [45], edge density [71] and higher intersection density [16] were associated with an increase in the number of days exceeding the air quality index for PM2.5. The findings of our study are partially, consistent with these results, but expand on the importance of the interaction among the number of patches, patch size and isolation, and the street design profiles. Specifically, the model including only the landscape profiles showed that the proximate stones profile, contiguous large inkblots and proximate inkblots profiles were associated with higher air pollution compared to the scattered pixels profile. After adjusting for the street design profiles, this pattern was preserved, although not significant. Our study also indicated that profiles with moderate to high street connectivity and moderate street directness (*semi-hyperbolic grid* and *spiderweb* profiles) were associated with high air pollution. Together, moderate directness of streets and high street connectivity could represent proxy indicators for walkability that are also related with lower speed and increased stop and go traffic, which in turn increases the levels of PM2.5 [16]. These results could be useful for the 12 Latin American cities in the C40 initiative for healthier and sustainable cities which belong to the *spiderweb* profile characterized by higher street connectivity, shorter streets and moderate directness [72].

Our findings should be interpreted considering the following limitations. First, there is currently no "perfect" (i.e., highly accurate) spatial mapping product characterizing the built environment at the national level and broad regional scales. However, our metrics are using some of the best available products, such as GUF representing higher-resolution urban footprints circa 2012, and we used a network research toolkit for modeling and analyzing large samples of street networks as nonplanar multigraphs [54]. Urban and public health researchers should proposed common indicators that are comparable across studies and that could be used in urban health studies for policy action. Second, there are differences in the measurement years of the metrics used in this paper and they represent a cross-sectional snapshot of the social determinants of health, the built environment, and its configuration, which does not capture the changing nature of cities nor their differing rates of change. However, SALURBAL includes the largest available sample of cities from Latin America with comparable measurements. Third, we did not include variables reflecting mix-land use and informal settlements due to limitations of available data.

Despite these limitations, our study represents an important step towards understanding built environment typologies across Latin American cities. Future work could capitalize on this effort to incorporate rates of change and spatio-temporal transformations of cities into such profiling. This approach would require the development of higher-accuracy urban land cover/land use maps at greater temporal frequencies, which remains an important need for many global cities. Our finite mixture modeling approach could be limited by a lack of an absolute measure of fit that made us rely on relative measures such as goodness-of-fit, classification, adherence to the hypothesis, and interpretability of profiles [58]. However, FMM has several advantages over other clustering methods as it is a parametric modeling approach that

could be potentially applied to cities outside of our sample to predict their underlying profiles. Moreover, it allows for the inclusion of control variables as predictors of profiles allowing for profiles to be used as predictor variables in other studies of health outcomes [73].

To conclude, this study demonstrates the heterogeneity of urban landscape and street design profiles in a large sample of Latin American cities. We show that urban typologies can be estimated using the spatial configuration of the built environment from a morphologic perspective and those typologies are also associated differently with measures of socioeconomic well-being and environmental quality. Our inclusion of a data dashboard allows researchers and practitioners to explore their cities in comparison to others in their country and elsewhere in the region. The profiles identified in our paper can be useful to describe current urban environments and assess their association with health outcomes and sustainability. Additionally, the profiles can be used as indicators when evaluating interventions that can alter the trajectory of cities towards being healthier, less unequal, and more sustainable. At the global level, our results could also be useful for The New Urban Agenda and the SDG 11 of Sustainable Cities & Communities and the incorporation of the 'Health in All Policies' HiAP framework.

## Supporting information

**S1 Table. Urban landscape and street design descriptives for Latin American countries.**
(DOCX)

**S2 Table. City profiles and conditional probabilities.**
(DOCX)

**S3 Table. Type of colonies and foundation year for 70 cities with highest conditional probabilities in profiles.**
(DOCX)

**S1 Fig. Elbow method selection for determining the number of latent classes.** A 1.5% of decrease in the Bayesian Information Criteria (BIC) function was used as stop criteria to determine the number of latent classes in each dimension. RE corresponds to relative entropy and PDBIC corresponds to the percentage of decreasing in BIC.
(DOCX)

**S2 Fig. Distribution of conditional probabilities of the cities to belong to each profile.** Distribution of the conditional probabilities of 370 cities is plotted in bars, where bigger bars reflect a higher proportion of cities. a) urban landscape profiles and b) street design profiles.
(DOCX)

**S1 Appendix. Urban landscape and street design descriptives for Latin American countries.**
(DOCX)

**S2 Appendix. Elbow method selection for determining the number of latent classes.**
(DOCX)

**S3 Appendix. Distribution of conditional probabilities of the cities to belong to each profile.**
(DOCX)

**S4 Appendix. City profiles and conditional probabilities.**
(DOCX)

**S5 Appendix. Type of colonies and foundation year for 70 cities with highest conditional probabilities in profiles.**
(DOCX)

## Acknowledgments

Members of the SALURBAL Group contributed to the overall conduct of the study as well as to data collection, data processing and data harmonization for this paper. The full SALURBAL group includes: Marcio Alazraqui, Hugo Spinelli, Carlos Guevel, Vanessa Di Cecco, Andres Trotta, Carlos Leveau, Adrián Santoro, Damián Herkovits: National University of Lanus, Buenos Aires, Argentina; Nelson Gouveia: Universidad de São Paulo, São Paulo, Brazil. Mauricio Barreto, Gervásio Santos: Oswaldo Cruz Foundation, Salvador Bahia, Brazil; Leticia Cardoso, Mariana Carvalho de Menezes, Maria de Fatima de Pina: Oswaldo Cruz Foundation, Rio de Janeiro, Brazil; Waleska Teixeira Caiaffa, Amélia Augusta de Lima Friche, Amanda Cristina de Souza Andrade, Universidade Federal de Minas Gerais, Belo Horizonte, Brazil; Patricia Frenz, Tania Alfaro, Cynthia Córdova, Pablo Ruiz, Mauricio Fuentes: School of Public Health, University of Chile, Santiago, Chile. Alejandra Vives Vergara, Alejandro Salazar, Andrea Cortinez-O'Ryan, Cristián Schmitt, Francisca Gonzalez, Fernando Baeza, Flavia Angelini: Department of Public Health, Pontificia Universidad Católica de Chile, Santiago, Chile; Olga Lucía Sarmiento, Diana Higuera, Maria Alejandra Wilches, Catalina González: School of Medicine, Universidad de los Andes, Bogotá, Colombia; Felipe Montes, Andres F. Useche, Oscar Guaje, Ana Maria Jaramillo, Luis Angel Guzmán: School of Engineering, Universidad de los Andes, Bogotá, Colombia. Philipp Hessel, Diego Lucumi: School of Government, Universidad de los Andes, Bogotá, Colombia; Jose David Meisel: Universidad de Ibagué, Ibagué, Colombia. Eliana Martinez: Universidad de Antioquia, Medellín, Colombia; María F. Kroker-Lobos, Manuel Ramirez-Zea, Kevin Martinez Folgar: INCAP Research Center for the Prevention of Chronic Diseases (CIIPEC), Institute of Nutrition of Central America and Panama (INCAP), Guatemala City, Guatemala; Tonatiuh Barrientos-Gutierrez, Carolina Perez-Ferrer, Javier Prado-Galbarro, Filipa de Castro, Rosalba Rojas-Martínez: Instituto Nacional de Salud Pública, Mexico City, Mexico; J. Jaime Miranda, Akram Hernández Vásquez, Francisco Diez-Canseco: School of Medicine, Universidad Peruana Cayetano Heredia, Lima, Peru; Ross Hammond: Brookings Institute, Washington, D.C., USA; Daniel Rodriguez, Iryna Dronova, Xize Wang, Mika Moran: Department of City and Regional Planning, the University of California Berkeley, USA; Peter Hovmand: Washington University in St Louis, St. Louis, Missouri, USA; Ricardo Jordán Fuchs, Juliet Braslow: Economic Commission for Latin America and the Caribbean (ECLAC); Jose Siri: United Nations University—International Institute for Global Health (UNU-IIGH); Ana Diez Roux, Amy Auchincloss, Usama Bilal, Brent Langellier, Gina Lovasi, Leslie McClure, Yvonne Michael, Kari Moore, Harrison Quick, D. Alex Quistberg, Brisa N. Sanchez, Ivana Stankov, Jose Tapia Granados: Dornsife School of Public Health, Drexel University, Philadelphia, Pennsylvania, USA.

We acknowledge Google Maps/Google Earth for the use of their content. Google expressly disclaims the accuracy, adequacy, or completeness of any data and shall not be liable for any errors, omissions or other defects in, delays or interruptions in such data, or any actions taken in reliance there on. All the figures were created by the researchers of this paper.

## Author Contributions

**Conceptualization:** Andrés F. Useche, Iryna Dronova, Felipe Montes, Ivana Stankov, Usama Bilal, Xize Wang, Fabian Peña, D. Alex Quistberg, John A. Guerra-Gomez.

**Data curation:** Andrés F. Useche, D. Alex Quistberg.

**Formal analysis:** Andrés F. Useche, Oscar Guaje, Felipe Montes.

**Funding acquisition:** Olga L. Sarmiento, D. Alex Quistberg, Ana V. Diez Roux.

**Investigation:** Olga L. Sarmiento, Andrés F. Useche, Daniel A. Rodriguez, Iryna Dronova, Oscar Guaje, Felipe Montes, Ivana Stankov, Maria Alejandra Wilches, Usama Bilal, Xize Wang, Luis A. Guzmán, Fabian Peña, John A. Guerra-Gomez, Ana V. Diez Roux.

**Methodology:** Olga L. Sarmiento, Andrés F. Useche, Daniel A. Rodriguez, Iryna Dronova, Usama Bilal.

**Project administration:** Olga L. Sarmiento, Ana V. Diez Roux.

**Resources:** Olga L. Sarmiento, Ana V. Diez Roux.

**Software:** Andrés F. Useche, Oscar Guaje, Felipe Montes, Maria Alejandra Wilches, Xize Wang.

**Supervision:** Olga L. Sarmiento, Daniel A. Rodriguez, Iryna Dronova, Ivana Stankov, Usama Bilal, Ana V. Diez Roux.

**Validation:** Olga L. Sarmiento, Andrés F. Useche, Daniel A. Rodriguez, Iryna Dronova, Oscar Guaje, Felipe Montes, Ivana Stankov, Maria Alejandra Wilches, Usama Bilal, Xize Wang, Luis A. Guzmán, Fabian Peña, D. Alex Quistberg, John A. Guerra-Gomez, Ana V. Diez Roux.

**Visualization:** Andrés F. Useche, Oscar Guaje, Felipe Montes, Fabian Peña, John A. Guerra-Gomez.

**Writing – original draft:** Olga L. Sarmiento, Andrés F. Useche, Daniel A. Rodriguez, Iryna Dronova, Oscar Guaje, Felipe Montes, Ivana Stankov, Maria Alejandra Wilches, Usama Bilal, Xize Wang, Luis A. Guzmán, Fabian Peña, D. Alex Quistberg, John A. Guerra-Gomez, Ana V. Diez Roux.

**Writing – review & editing:** Olga L. Sarmiento, Andrés F. Useche, Daniel A. Rodriguez, Iryna Dronova, Oscar Guaje, Felipe Montes, Ivana Stankov, Maria Alejandra Wilches, Usama Bilal, Xize Wang, Luis A. Guzmán, Fabian Peña, D. Alex Quistberg, John A. Guerra-Gomez, Ana V. Diez Roux.

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
