## [Decision Letter · Decision Letter 0]

28 Apr 2021

PONE-D-20-18493

Built Environment Profiles for Latin American Urban Settings: The SALURBAL study

PLOS ONE

Dear Dr. Sarmiento,

Firstly, please accept my sincere apologies for the time it has taken for us to get a first decision to you. We have now completed the peer review process, and your manuscript has been assessed by two external experts, whose reports are appended to this letter. After careful consideration of these reports, we feel your submission has merit but does not fully meet PLOS ONE’s publication criteria as it currently stands. 

The reviewers raised a number of points regarding specific aspects of the methodology and motivation/goals for your study. Therefore, we invite you to submit a revised version of the manuscript that addresses the points raised during the review process. 

We look forward to receiving your revised manuscript. Again, I apologise for the extended delay in processing your submission - if you have concerns about the progress of your submission in future then do feel free to contact me directly at jdonlan@plos.org.

Kind regards,

Dr Joseph Donlan

Senior Editor

PLOS ONE

Journal Requirements:

PLOS requires an ORCID iD for the corresponding author in Editorial Manager on papers submitted after December 6th, 2016. Please ensure that you have an ORCID iD and that it is validated in Editorial Manager. To do this, go to ‘Update my Information’ (in the upper left-hand corner of the main menu), and click on the Fetch/Validate link next to the ORCID field. This will take you to the ORCID site and allow you to create a new iD or authenticate a pre-existing iD in Editorial Manager. Please see the following video for instructions on linking an ORCID iD to your Editorial Manager account: https://www.youtube.com/watch?v=_xcclfuvtxQ

The Salud Urbana en América Latina (SALURBAL)/ Urban Health in Latin America project is funded by the Wellcome Trust [205177/Z/16/Z]. More information about the project can be found at www.lacurbanhealth.org. UB was partially supported by the Office of the Director of the National Institutes of Health under award number DP5OD26429.  Mention of trade names, commercial practices, or organizations does not imply endorsement by the authors, the institutions where the authors work, or funding entities. The funders had no role in study design, data collection and analysis, decision to publish, or preparation of the manuscript.

Reviewers' comments:

Reviewer's Responses to Questions

**Comments to the Author**

1. Is the manuscript technically sound, and do the data support the conclusions?

Reviewer #1: Yes

Reviewer #2: Yes

2. Has the statistical analysis been performed appropriately and rigorously? 

Reviewer #1: Yes

Reviewer #2: Yes

3. Have the authors made all data underlying the findings in their manuscript fully available?

Reviewer #1: Yes

Reviewer #2: Yes

4. Is the manuscript presented in an intelligible fashion and written in standard English?

Reviewer #1: Yes

Reviewer #2: Yes

5. Review Comments to the Author

Reviewer #1: The authors identify city profiles based on the built landscape and street design characteristics of cities in Latin America and evaluate the associations of city profiles with social determinants of health and air pollution. The manuscript is well-structured and contributes to studies related to morphological classifications of cities/urban areas and their associations with social aspects of health determinants. In addition, the manuscript also contributes to the literature on morphological classification of cities in Latin American context.

Some specific comments and suggestions are listed below:

1. A literature review of related past studies is missing and it is important to justify the methods used, to identify the gaps and to highlight the originality and contribution of this study to literature. More particularly, past studies using different pattern analysis to identify various profiles of cities should be highlighted. One such study is: Southworth, M., & Owens, P. M. (1993). The evolving metropolis: Studies of community, neighborhood, and street form at the urban edge. Journal of the American Planning Association, 59(3), 271-287. A more recent study is Peponis, J., Allen, D., Haynie, D., Scoppa, M., & Zhang, Z. (2007). Measuring the configuration of street networks: the spatial profiles of 118 urban areas in the 12 most populated metropolitan regions in the US. These are based on the classification of street connectivity of urban areas. What are the key metrics used previously? More regarding different measures should be cited. In addition, studies investigating the relationship between these patterns and health outcomes need to be cited too. Many previous studies are cited in the Discussion, but these should have been introduced in the Lit Rev.

2. In Table 1, why are no definitions provided for certain metrics used: i.e. directness? These already have been defined in the literature and indeed the authors define them later in the text.

3. Line 198: why is the threshold set to 25?

4. Table 2 shows the correlations of individual measures with social aspects of health determinants. I believe these need some discussion/explanation. For example, to me it is interesting and unexpected why intersection density is negatively correlated with air pollution whereas street density is positively correlated with the same measure.

5. Line 331: should be edited as “95% confidence interval” to be consistent with the note on Table 6.

6. The manuscript can be stronger if, in the Discussion, the authors discuss briefly the policy/planning implications of their findings, particularly the associations of urban landscape and street design with health aspects. Can any suggestions for urban planners be drawn based on the findings?

7. There are a couple sentences that need to be revised grammatically.

line 467: “of the fragmentation an density of…” and?

line 509: “Latin American…” should be Latin America

Reviewer #2: This is an interesting study describing landscape and street patterns across Latin American cities. I believe it requires some revision prior to be acceptable for publication. In particular, it needs a clearer motivation and takeaway message, as well as more clarity in some of the methods.

Throughout: change "OpenStreetMaps" to "OpenStreetMap"

Introduction: I'd like to see a clearer motivation here for why these research questions are important. What is the value of generating this new knowledge? Who will use it and how will they do things differently after reading your paper than they would have beforehand?

Where do the study site boundary geometries come from? Is that the AUE specifically? Worth making that explicit.

Pages 5-9: For both the urban landscape domain and street design domain, you mention that you selected a few metrics to measure different aspects. But how did you select these specific metrics? Why these? I'd like to see more theoretical development here or in the background to make a case for why these specific dimensions capture the most important physical aspects of these cities for your study. More motivation for their selection.

Page 10: you mention that the socioeconomic variables come from the nations' census bureaus. Do these variables have the same meaning, coverage, interpretation, etc across the various nations? They are directly comparable with one another?

Page 10: regarding the particulate matter concentration raster, it strikes me that it may be better to population-weight the mean so that you measure exposure. For example, if half the pollution is concentrated in an industrial area where very little of the population is exposed to it, an unweighted mean may not give an accurate snapshot.

Page 11: you briefly introduce what FMM is capable of, but it may be useful to briefly explain what it does and how it does it for the general readership.

Pages 11-12: please provide mathematical equations for your models to make their specification clearer

Results section: rather than just presenting who was lowest/highest on different indicators, it would help to briefly explain/interpret why they were lowest or highest. What are the unique characteristics or histories of these places to explain their extreme values? Also, consider presenting a table to summarize some of these values, or perhaps a figure showing box plots to demonstrate their distributions.

The visualization tool on your web site is a nice feature.

Discussion section: I feel like the paper concludes without a takeaway message right now. This is a correlational/descriptive study, but I'd like to see more motivation for why these descriptions matter. How can policymakers use these findings? What new theoretical knowledge do they generate? Why is that new knowledge urgently needed to understand and improve urban living?

The last paragraph uses causal language ("determining", "influence") that does not appear to be warranted as the study did not employ a causal research design such as a quasi-experimental framework.

6. PLOS authors have the option to publish the peer review history of their article (what does this mean?). If published, this will include your full peer review and any attached files.

Reviewer #1: No

Reviewer #2: No

---

## [Author Response · Author response to Decision Letter 0]

27 Jul 2021

Dear reviewers,

We greatly appreciate your feedback and constructive comments. We addressed each comment by thoroughly revising our manuscript. The actions taken to respond to each comment are detailed in the following text. All the changes are also highlighted in blue within the word text of the manuscript and we provided page and line numbers where you can find those changes. We included the word file with the track changes and without track changes

Reviewer 1

Summary of work

 “The authors identify city profiles based on the built landscape and street design characteristics of cities in Latin America and evaluate the associations of city profiles with social determinants of health and air pollution. The manuscript is well-structured and contributes to studies related to morphological classifications of cities/urban areas and their associations with social aspects of health determinants. In addition, the manuscript also contributes to the literature on morphological classification of cities in Latin American context”.

Answer:

We greatly appreciate the encouraging comments of the reviewer.

Remarks

 “A literature review of related past studies is missing, and it is important to justify the methods used, to identify the gaps and to highlight the originality and contribution of this study to literature. More particularly, past studies using different pattern analysis to identify various profiles of cities should be highlighted. One such study is: Southworth, M., & Owens, P. M. (1993). The evolving metropolis: Studies of community, neighborhood, and street form at the urban edge. Journal of the American Planning Association, 59(3), 271-287. A more recent study is Peponis, J., Allen, D., Haynie, D., Scoppa, M., & Zhang, Z. (2007). Measuring the configuration of street networks: the spatial profiles of 118 urban areas in the 12 most populated metropolitan regions in the US. These are based on the classification of street connectivity of urban areas. What are the key metrics used previously? More regarding different measures should be cited. In addition, studies investigating the relationship between these patterns and health outcomes need to be cited too. Many previous studies are cited in the Discussion, but these should have been introduced in the Lit Rev”.

Answer:

We appreciated the recommendation of the reviewer. 

We included the following changes in the manuscript: 1) we expanded the introduction with a literature review including the literature on the built environment (urban landscape and street design metrics) in previous studies in Latin America; 2) These metrics were compared with studies in other countries; 3) we included the literature review of the association of built environment and social determinants of health with emphasis on those conducted in Latin America, 3) we included a literature review of urban profiles, 4) we included a paragraph on the innovation of the study and the potential use of this results for inform global and local policies, 5) lastly, we added your suggested references regarding profiles and typologies. 

Changes can be found on: 

1) Introduction section: pages: 3-7; lines 77-177

2) Discussion section: pages: 33-39; lines 605-774

 “In Table 1, why are no definitions provided for certain metrics used: i.e. directness? These already have been defined in the literature and indeed the authors define them later in the text”.

Answer:

We completed table 1 with the missing definitions for the number of urban patches, patch density, and circuity average. 

Changes can be found on: 

Page 9: Line: 223

 “Line 198: why is the threshold set to 25?”

Answer:

The threshold was set according to the metrics in the SALURBAL project. This variable has been used for characterizing variability and predictors of infant mortality in urban settings. Specifically, 25 years is the minimum age standard for measuring adult educational attainment defined by UNESCO and OECD. This age is used because it assumes that most adults will have already completed basic, high school, and college education in most countries. To clarify this cut point we included the following reference:

Ortigoza AF, Tapia Granados JA, Miranda JJ, et al. Characterising variability and predictors of infant mortality in urban settings: findings from 286 Latin American cities. J Epidemiol Community Health 2021;75:264-270. 

Changes can be found on: 

Page: 14 ; Lines: 310-315

 “Table 2 shows the correlations of individual measures with social aspects of health determinants. I believe these need some discussion/explanation. For example, to me it is interesting and unexpected why intersection density is negatively correlated with air pollution whereas street density is positively correlated with the same measure”.

Answer:

We appreciate the comment we want to clarify that the table had an error. When we made the table, one column was shifted and that was the reason why some of the correlations seem odd. We fixed this error, and table 2 now is updated. In the discussion section, we also expanded the hypothesis regarding the associations of air pollution with urban landscape profiles and metrics and street design profiles and metrics. 

Changes can be found on: 

Page: 22; Lines: 471-482

Page: 23 for table 2

 “Line 331: should be edited as “95% confidence interval” to be consistent with the note on Table 6”.

Answer:

According to the recommendation of the reviewer, we changed the confidence interval to 95% confidence interval.

The corrected table footnote is: The 95% confidence intervals are shown in square brackets. 

Change can be found on: 

Page: 30; line: 580

 “The manuscript can be stronger if, in the Discussion, the authors discuss briefly the policy/planning implications of their findings, particularly the associations of urban landscape and street design with health aspects. Can any suggestions for urban planners be drawn based on the findings?”

Answer: 

According to the reviewers, suggestions we included the policy recommendations in the discussion section.

 “There are a couple sentences that need to be revised grammatically.

line 467: “of the fragmentation an density of…” and?

line 509: “Latin American…” should be Latin America”

Answer:

We changed the sentences to be grammatically correct.  

Reviewer 2

Summary of work

 “This is an interesting study describing landscape and street patterns across Latin American cities. I believe it requires some revision prior to be acceptable for publication. In particular, it needs a clearer motivation and takeaway message, as well as more clarity in some of the methods”.

Answer:

We included the following changes in the manuscript: 1) we expanded the introduction with a literature review including the literature on the built environment (urban landscape and street design metrics) in previous studies in Latin America; 2) These metrics were compared with studies in other countries; 3) in the methodology we expanded the justification of the use of our metrics; 4) we included the literature review of the association of built environment and social determinants of health with emphasis on those conducted in Latin America, 5) we included a literature review of urban profiles, and 6) we included a paragraph on the innovation of the study and the potential use of this results for inform global and local policies. 

Changes can be found on the following sections: 

1) Introduction section: pages: 3-7; lines 77-177

2) Discussion section: pages: 33-39; lines 605-774

Remarks

 “Throughout: change "OpenStreetMaps" to "OpenStreetMap"”.

Answer:

According to the recommendation of the reviewer, we corrected the word OpenStreetMap.

Changes can be found on lines 160, 285 and 286 

 “Introduction: I'd like to see a clearer motivation here for why these research questions are important. What is the value of generating this new knowledge? Who will use it and how will they do things differently after reading your paper than they would have beforehand?”

Answer:

We included the following changes in the manuscript: 1) we expanded the introduction with a literature review including the literature on the built environment (urban landscape and street design metrics) in previous studies in Latin America; 2) These metrics were compared with studies in other countries; 3) we included the literature review of the association of built environment and social determinants of health with emphasis on those conducted in Latin America, 4) we included a literature review of urban profiles, and 5) we included a paragraph on the innovation of the study and the potential use of this results for inform global and local policies. 

Changes can be found on the following sections: 

1) Introduction section: pages: 3-7; lines 77-177

2) Discussion section: pages: 33-39; lines 605-774

 “Where do the study site boundary geometries come from? Is that the AUE specifically? Worth making that explicit”.

Answer:

Cities were included in SALURBAL based on city size >=100,000 as determined by the AUE and official census data and boundaries were based on official administrative boundaries of subcity units in each country that encompassed the visual urban extent of each city. We obtained administrative boundaries from official country sources and overlayed those into satellite imagery of each city to determine the urban extent.

Changes can be found on the methods section: page 8: Lines: 193-197 

 “Pages 5-9: For both the urban landscape domain and street design domain, you mention that you selected a few metrics to measure different aspects. But how did you select these specific metrics? Why these? I'd like to see more theoretical development here or in the background to make a case for why these specific dimensions capture the most important physical aspects of these cities for your study. More motivation for their selection”.

Answer:

The three landscape subdomains, together with area and population, encompass attributes that can be used to characterize urban development. Previous studies developed various strategies to measure urban form in particular, which may differ among application objectives. Furthermore, a number of spatial metrics have been proposed to assess landscape configuration based on geometry and spatial relationships among discrete units, or patches. To more effectively identify typologies of urban morphology landscape structure for SALURBAL cities in this study, we chose a parsimonious set of landscape metrics based the principles of strength, universality and consistency - i.e., metrics that represent different components of landscape structure such as size, shape, texture, and contiguity, and that, by design, would not be expected to strongly correlate with each other.

We expanded the definition and rationale for using these metrics. 

Changes can be found on the methods section: page 8: Lines: 219-228

Within the street design domain, we selected five metrics that represent the subdomains of street connectivity, street length, and directness. We selected these metrics that capture the cities’ general street network structure and have been used in multiple studies allowing comparability

We expanded the definition and rationale for using these metrics. 

Changes can be found on the methods section: page 13: Lines: 291-294

 “Page 10: you mention that the socioeconomic variables come from the nations' census bureaus. Do these variables have the same meaning, coverage, interpretation, etc across the various nations? They are directly comparable with one another?”

Answer:

These metrics were standardized among countries according to common thresholds. As such they are comparable. We clarified this in the manuscript and provided a reference.

Ortigoza AF, Tapia Granados JA, Miranda JJ, et al. Characterising variability and predictors of infant mortality in urban settings: findings from 286 Latin American cities. J Epidemiol Community Health 2021;75:264-270.

Changes can be found on the methods section: page 14: Lines: 321-324

 “Page 10: regarding the particulate matter concentration raster, it strikes me that it may be better to population-weight the mean so that you measure exposure. For example, if half the pollution is concentrated in an industrial area where very little of the population is exposed to it, an unweighted mean may not give an accurate snapshot”.

Answer:

We thank the reviewer for this insightful comment. While we agree there are many situations where a population-weighted air pollution metric would be ideal, given the emphasis of the paper on urban landscape profiles of cities as a whole, we feel that it is appropriate to use an unweighted air pollution metric that represents mean characteristics within the same geographic area as the urban landscape variables in the analysis. In examining city profiles, we do not distinguish between built environment characteristics in residential, industrial, or green spaces within each city because the explicit purpose is to look at each city as a whole. The urban characteristics (for example, street connectivity) used to create the city profiles in the analysis represent the entire geographic area within the political boundaries of the city. Given the specific urban characteristics we used to develop profiles of urban landscape and street design, we expect that associations between these built environment profiles and air pollution are largely driven by city-level differences related to road networks and vehicular traffic profiles. Within each city, traffic-related pollution may or may not have a strong spatial correlation with residential areas, and these residential areas would, of course, be given more weight with a population-weighted air pollution metric. We believe that for our aim of determining associations between different built environment city profiles and differences in city-level air pollution concentrations, the most accurate approach is to use an unweighted air pollution metric that spatially matches the urban landscape and street design characteristics.

Clarifications can be found on the methods section: page:15 Lines: 344-347

 “Page 11: you briefly introduce what FMM is capable of, but it may be useful to briefly explain what it does and how it does it for the general readership”.

Answer:

We have expanded the description of the FMM including the following sentences: “FMM uses a mixture of distributions to model the heterogeneity of group structures. This methodology is defined in terms of the measurements and their probability density functions. Let Y_1,…,Y_n the realizations of the p measurements in the sample Y [28]. The probability density function f(y_j ) can be described as:

f(y_j )=∑_(i=1)^g▒〖π_i*f_i (y_j)〗

where f_i (y_j) denotes the component densities of the mixture, g is the number of components of the finite mixture distribution, π_i are called the weights or conditional probabilities to belong to each component. These weights are nonnegative quantities that sum one:

■(■(0≤π_i≤1&(i=1,…,g))@∑_(i=1)^g▒π_i =1)

The value of g and the conditional probabilities π_i are unknown and must be inferred from the data [28]. The selection of the number of components was conducted using as criteria the Bayesian Information Criteria (BIC) and the entropy. The BIC was used to evaluate the goodness of fit of the model and the entropy was used to evaluate a good classification of the observations within latent constructs. The observations were assigned to the latent construct with the highest conditional probability”. 

Changes can be found on the methods section: pages 15-16: Lines: 355-366

 “Pages 11-12: please provide mathematical equations for your models to make their specification clearer”.

Answer:

We provided the mathematical equations

 The first level can be described as:

■(■((1)&Y_ij=b_0j+b_1j X_ij+ε_ij )&,ε_ij~N(0,σ^2 ) )

where Y_ij is the outcome for the i^th individual in the j^th group, b_0j is the intercept for the j^th group, b_1j is the coefficient for the matrix of variables for the j^th group, X_ij is the variables matrix for the i^th individual in the j^th group and ε_ij is the i^th individual error in the j^th group. The outcome Y_ij correspond to the social determinants of health and air pollution. The variables used to define the X_ij matrix were the urban landscape and street design profiles.

The second level can be described as:

■((2)&■(■(b_0j=γ_00+γ_01 C_j+U_0j&U_0j~N(0,τ_00))@■(b_1j=γ_10+γ_11 C_j+U_1j&U_1j~N(0,τ_11))))

where γ_00 is the common intercept across groups, γ_10 is the common slope associated with X_ij across groups, γ_01 is the change in the intercept per unit change C_j, γ_11 is the change in the slope per unit change C_j, U_0j is the unique deviation of the intercept of each group from the overall intercept, U_1j is the unique deviation of the slope within each group from the overall slope, τ_00 is the variance of the group specific intercept and τ_11 is the variance of the group specific slope.”

Changes can be found on the methods section: pages 17-18: Lines: 394-410

 Results section: rather than just presenting who was lowest/highest on different indicators, it would help to briefly explain/interpret why they were lowest or highest. What are the unique characteristics or histories of these places to explain their extreme values? Also, consider presenting a table to summarize some of these values, or perhaps a figure showing box plots to demonstrate their distributions.

Answer:

According to your recommendations, we included examples of the cities within each profile underscoring those cities with the highest conditional probability. We included a supplementary table including indicators of type colony, population density and area that could explain in part the configuration of the city. Future studies should take into account our results in the context of the history of each city, transport and urban planning policies, and economic development. 

We also included two figures (figure 1 and figure 2) with histograms showing the scale and dispersion of each metric for all studied cities. We did not include box plots due to different scales and the large dispersion among the metrics, but included the histograms for helping the visualization of the results. 

In addition, we want to point out that these suggested tables can be complemented with the information of the dashboard.

Changes can be found on the results section: pages 20 and 21: Lines: 454 and 479

In addition we included a Table of supplementary material including indicators of type colony, population density and area

 “The visualization tool on your web site is a nice feature”.

Answer:

Thank you for the comment.

 “Discussion section: I feel like the paper concludes without a takeaway message right now. This is a correlational/descriptive study, but I'd like to see more motivation for why these descriptions matter. How can policymakers use these findings? What new theoretical knowledge do they generate? Why is that new knowledge urgently needed to understand and improve urban living?”

Answer:

According to the reviewers comment we expanded the policy and research implications of our study.

Our study characterizes the urban landscape and street design of 370 cities in Latin America and their association with social and environment determinants of health We identified four profiles for the urban landscape and four for street design. The characteristics and composition of the profiles highlight the heterogeneity and complexity of cities in the region. When examining associations with social and environmental determinants of health, we did not find a best “potentially healthier” profile. Profiles of cities including higher street connectivity with moderate directed streets exhibit better socioeconomic conditions, but more air pollution; while profiles including more fragmentation, less isolation and more complex shape were associated with more traffic congestion.

Prior applications of profiles have emphasized their descriptive nature and their use as triggers of policy change, either to address concerns such as fragmentation and isolation, or as an aspirational vision to be achieved. Yet, the profiles in this study were not consistently and unequivocally associated with positive environmental and social outcomes. They underscore the tradeoffs often present in areas of rapid growth and economic need: economic development and productivity improving housing and social development but beget air pollution and congestion. Together, these profiles provide a novel conceptual perspective to understanding urban areas. Especially for Latin America, the inclusion of cities with different sizes sheds new light on the organization of cities and can serve as indicators to guide decisions about urban planning, transport and health. 

Additionally, the profiles can be used as indicators when evaluating interventions that can alter the trajectory of cities towards being healthier, less unequal, and more sustainable. At the global level, our results could also be useful for The New Urban Agenda and the SDG 11 of Sustainable Cities & Communities and the incorporation of the ‘Health in All Policies’ approach.

Changes can be found on the discussion section: pages 33-34: Lines: 620-640

 “The last paragraph uses causal language ("determining", "influence") that does not appear to be warranted as the study did not employ a causal research design such as a quasi-experimental framework”.

Answer:

Following the reviewer’s suggestion, we adjusted the language for avoiding confusion about causality conclusions of cities towards more healthy, less unequal, and sustainable outcomes.

---

## [Editor Report · Decision Letter 1]

7 Sep 2021

Built Environment Profiles for Latin American Urban Settings: The SALURBAL study

PONE-D-20-18493R1

Dear Dr. Sarmiento,

We’re pleased to inform you that your manuscript has been judged scientifically suitable for publication and will be formally accepted for publication once it meets all outstanding technical requirements. As disclosure, note that I was one of the reviewers of the initial submission.

Kind regards,

Geoff Boeing

Guest Editor

PLOS ONE

---

## [Editor Report · Acceptance letter]

14 Oct 2021

PONE-D-20-18493R1 

Built Environment Profiles for Latin American Urban Settings: The SALURBAL study 

Dear Dr. Sarmiento:

I'm pleased to inform you that your manuscript has been deemed suitable for publication in PLOS ONE. Congratulations! Your manuscript is now with our production department. 

Kind regards, 

on behalf of

Dr. Geoff Boeing 

Guest Editor

PLOS ONE